# A Novel Piggyback Strategy for mRNA Delivery Exploiting Adenovirus Entry Biology

**DOI:** 10.3390/v14102169

**Published:** 2022-09-30

**Authors:** Myungeun Lee, Paul J. Rice-Boucher, Logan Thrasher Collins, Ernst Wagner, Lorenzo Aulisa, Jeffrey Hughes, David T. Curiel

**Affiliations:** 1Division of Cancer Biology, Department of Radiation Oncology, School of Medicine, Washington University in St. Louis, St. Louis, MO 63110, USA; 2Department of Biomedical Engineering, McKelvey School of Engineering, Washington University in Saint Louis, St. Louis, MO 63130, USA; 3Department of Chemistry and Pharmacy, Ludwig-Maximilians-University (LMU), 81377 Munich, Germany; 4GreenLight Biosciences, Inc., 200 Boston Ave. #3100, Medford, MA 02155, USA; 5Biologic Therapeutics Center, Department of Radiation Oncology, School of Medicine, Washington University in St. Louis, St. Louis, MO 63110, USA

**Keywords:** targeted adenoviral vectors (Ad), streptavidin-polylysine (STAVpLys), messenger Ribonucleic Acid (mRNA)

## Abstract

Molecular therapies exploiting mRNA vectors embody enormous potential, as evidenced by the utility of this technology for the context of the COVID-19 pandemic. Nonetheless, broad implementation of these promising strategies has been restricted by the limited repertoires of delivery vehicles capable of mRNA transport. On this basis, we explored a strategy based on exploiting the well characterized entry biology of adenovirus. To this end, we studied an adenovirus-polylysine (AdpL) that embodied “piggyback” transport of the mRNA on the capsid exterior of adenovirus. We hypothesized that the efficient steps of Ad binding, receptor-mediated entry, and capsid-mediated endosome escape could provide an effective pathway for transport of mRNA to the cellular cytosol for transgene expression. Our studies confirmed that AdpL could mediate effective gene transfer of mRNA vectors in vitro and in vivo. Facets of this method may offer key utilities to actualize the promise of mRNA-based therapeutics.

## 1. Introduction

Nucleic acid vectors based upon mRNA have emerged as a transformative technology to address emerging pandemics. Indeed, the rapid development, and deployment, of mRNA vaccines for SARS-CoV-2 proved pivotal in limiting the most dire consequences of COVID-19 [1,2,3,4,5,6,7]. In this regard, the most salient advantage of the mRNA vector technology is the rapidity by which a specific vaccine can be derived for a new pandemic agent. Based upon this distinguishing characteristic, considerable efforts are currently being directed towards advancing the pharmacologic aspects of mRNA to improve its utility, especially for the context of emerging pandemic-driven threats.

Of note, the pandemic-driven advancement of mRNA vector technology has enabled consideration of applying this delivery approach for a broad range of disorders. In this regard, mRNA-based gene therapy strategies are currently being evaluated for classical inherited genetic disease targets. In addition, the pharmacology of mRNA-based gene delivery potentially provides a useful platform to approach a range of acquired disorders. This diversity of candidate applications has suggested that mRNA may provide truly revolutionary possibilities to realize molecular medicine cures.

To this point, however, the application of mRNA vaccines has been based upon direct delivery of the naked nucleic acid vector or complexing of the mRNA with lipid nanoparticles (LNP). These vector platforms, however, limit the delivery of mRNA to a very restricted repertoire of routes [1,8,9]. This consideration currently represents the major impediment to realizing the anticipated broad utilities of mRNA-based approaches for human disease.

In this regard, we have previously exploited adenovirus to facilitate delivery of DNA-based nucleic acid vectors [10,11]. In this “piggyback” approach, adenovirus is coupled to a polycation that provides the basis of electrostatic association with plasmid DNA vectors. This “adenovirus-polylysine” (AdpL) strategy was designed to exploit adenovirus to facilitate macromolecular transport in a novel manner that potentially embodied distinct advantages compared to conventional adenovirus vectors. These advantages included a greatly expanded packaging capacity for heterologous DNA transport as well as the transport of novel functional capacities in trans. The AdpL was capable of in vivo delivery to target tissue contexts not approachable with naked plasmid DNA alone [12,13]. In addition, its use for generation of a gene-modified cancer vaccine under GMP conditions with application in a clinical melanoma vaccine study has demonstrated pharmaceutical feasibility of the AdpL system [14].

Based upon these considerations, we evaluated the possibility of AdpL-based delivery of mRNA. Our studies here demonstrate highly efficient gene transfer of mRNA via the AdpL approach. Of note, the AdpL was capable of accomplishing gene transfer via mRNA in vivo. The novel “piggyback” gene delivery strategy of AdpL thereby represents a new method for delivery of mRNA that may enable useful practical strategies to realize the potential of this promising vector. These studies highlight again the gains that may accrue exploiting adenoviral entry biology to facilitate macromolecular transport.

## 2. Results

### 2.1. Design of a Novel Strategy for “Piggyback” Delivery of mRNA

We previously utilized a piggyback approach to accomplish gene delivery of plasmid DNA for gene therapy purposes [10,11]. In this design, the polycation polylysine (pLys) coupled to the adenovirus capsid provides the basis of electrostatic association with the negatively charged nucleic acid species. On this basis, we sought to explore the possible utility of this adenovirus-polylysine (AdpL) vector for delivery of mRNA. In our earlier studies we explored various methods for attachment of the polylysine binding moiety to the adenovirus capsid [15,16]. For current study, we exploited our previously method whereby streptavidin (STAV) conjugated polylysine (STAVpLys, molar ratio of STAV to pLys_250_ = 1:2) is linked with a biotin-labelled adenovirus vector (Figure 1) [10]. To assess the capacity of this system to transduce multiple cell and tissue types, we utilized an adenoviral vector based on human adenovirus serotype 5 (HAdV5) with fiber knob chimerism whereby the knob domain of the HAdV5 is replaced with the fiber knob of porcine adenovirus serotype 4. In our earlier studies we showed that the resulting chimeric adenovirus vector (AdPK4) possesses enhanced infectivity based on gene transfer that exploits glycan binding of target cells [17]. We hypothesized that the negatively charged mRNA would associate with the positively charged polylysine in a manner similar to plasmid DNA, and that the resulting AdpL-mRNA conjugate would facilitate delivery of the mRNA in a manner comparable to what we had observed for plasmid DNA.

### 2.2. Assessment of Adenovirus-Polylysine-Mediated mRNA Delivery

To validate our overall hypothesis of AdpL-mediated mRNA delivery, we first utilized a luciferase encoding mRNA and a GFP encoding AdPK4 adenoviral vector to access AdpL-mediated mRNA gene transfer. To test if mRNA complexation with the adenovirus capsid can specifically mediate gene transfer, we tested the AdpL-mRNA vector alongside essential controls to evaluate the contribution of each component of the system. We tested in CHO (Chinese Hamster Ovary) cells AdPK4 only, STAVpLys-mRNA only, non-biotin labelled AdPK4 plus STAVpLys-mRNA, or biotin-labelled AdPK4 plus STAVpLys-mRNA. These vector designs (AdpL containing 800 ng StAV and 880 ng pLys) were constituted with varying amounts of mRNA (0.375 µg–3.0 µg) and luciferase gene expression was analyzed forty-eight hours after virus infection (Figure 2A). These studies clearly validated gene transfer mediated via the AdpL vehicle. Of note, in the 0.375, 0.75, and 1.5 ug groups, the AdpL-mediated mRNA gene transfer was substantially greater than any of the control groups (Figure 2A). This observation demonstrates that a specific ratio of the polylysine-messenger nucleic acid complex (charge ratio of pLys to mRNA of 6, 3, or 1.5) corresponded with optimal gene transfer. Interestingly, in the higher mRNA groups we did not observe significant gene transfer, potentially due to incomplete mRNA binding and protection (Figure 2B). Furthermore, the non-biotin labelled AdPK4-STAVpLys-mRNA samples also showed some signs of luciferase expression, possibly due to non-specific interactions between the viral particles and mRNA fostering uptake. Based on the empirical results, we moved forward with the optimal AdpL-mRNA ratio for our further studies.

### 2.3. Polylysine-Mediated mRNA Conjugation Alters Adenovirus Tropism

Adenoviral vectors mediate gene delivery that is dictated by the tropism of the parental adenovirus species. To this end, various methods have been employed to expand adenovirus tropism towards the goal of optimized gene delivery [18,19,20,21,22]. Of note in this regard, polylysine can interact with a range of cellular membrane surface molecules [23,24]. When complexed to adenovirus, this polylysine binding could potentially alter the native tropism of the adenovirus infection for effective gene delivery. On this basis, we hypothesized that the polylysine component of the AdpL could potentially expand the range of cellular targets susceptible to adenovirus mediated gene delivery. For this study, we utilized the human embryonic cell line 293, which has ample expression of cell surface receptors for the porcine knob 4 binding domain of AdPK4. As control, we also utilized CHO, which is negative for the adenovirus porcine knob 4 binding domains. We infected each cell line with AdPK4.CMV.GFP, which expresses GFP from the viral DNA as a reporter, polylysine complexed luciferase encoding mRNA (STAVpLys-mRNA), or AdpL-mRNA. As expected, we observed that AdPK4.CMV.GFP effectively delivered the GFP reporter to 293 cells while the negative control CHO cells were resistant, and the polycation complexed mRNA (STAVpLys-mRNA) did not mediate any gene transfer. On the other hand, the AdpL-mRNA that embodied the AdPK4 component achieved highly efficient gene transfer of the GFP reporter in both 293 cells as well as the CHO cells (Figure 3). This observation suggests that, in the context of the AdpL-mRNA design, the polylysine component can contribute to vector tropism.

### 2.4. Adenovirus-Polylysine Co-Delivery of Two mRNAs

Delivery of mRNAs encoding separate proteins would be of strong utility, especially in the context of vaccine strategies where delivery of multiple antigens would be advantageous. We therefore sought to determine if the AdpL-mRNA system could successfully deliver two mRNAs. We complexed the two mRNAs with the AdpL at a one-to-one ratio and infected 293 cells. We observed reporter expression from each different mRNA in addition to the reporter encoded by the adenovirus components of the AdpL (Figure 4).

### 2.5. In Vivo Delivery of mRNA via Adenovirus-Polylysine Strategy

We next sought to determine if the AdpL vector was capable of achieving in vivo delivery of mRNA. We derived AdpL utilized the described GFP expressing AdPK4.CMV.GFP to enable tracking of in vivo gene transfer mediated by the adenovirus. An mRNA encoding a luciferase reporter was complexed to this AdpL vector and administered via the intramuscular route. The AdpL mediated mRNA transfer was evaluated via fluorescence imaging of the muscle (Figure 5). We found colocalized GFP and RFP signals, confirming in vivo gene delivery deriving exclusively from mRNA after AdpL mediated delivery.

## 3. Discussion

In this study, we demonstrated a novel vector strategy to accomplish delivery of gene expression- competent mRNA. We achieved this goal utilizing an approach whereby the mRNA is complexed to the adenovirus capsid exterior in a piggyback fashion. Critical to the functionality of the AdpL-mRNA vector are the functions provided by the adenovirus. Specifically, we hypothesize that the adenovirus in the provides efficient target cell binding, cellular entry, and escape from the endosome (Figure 6). This model is based on our previous work with HAdV5 and DNA, and we believe a similar effect is at play here, although the exact entry and escape steps of the AdPK4 vector need to be elucidated. In the aggregate, these adenovirus functionalities are successfully exploited to achieve efficient mRNA vector delivery.

Of note, the major vectors systems employed for mRNA delivery here-to-fore have been based upon liposomes or lipid nanoparticles (LNP) [3,4,25,26,27]. Despite the demonstrated utility of these vectors in various model systems, these vectors overall are limited in some critical ways. Firstly, liposomes and LNPs are relatively inefficient compared to viral vectors. In this regard, their lack of specific domains to accomplish the critical steps of target cell binding, entry, and endosome release all contribute to the relatively lower delivery efficiency. To this end, specific selection and directed engineering steps have been employed to design liposome and LNP vectors that embody these functionalities. Nonetheless, these engineering efforts remain largely empiric such that efficiency remains the major limiting factor in the employ of liposomes and LNPs.

The issue of efficiency is of even great relevance for the context of in vivo delivery. Of note, the overwhelming majority of published studies utilizing mRNA for vaccine applications employ an intramuscular delivery route [1,27]. This is owing to the fact that muscle seems to be uniquely susceptible to mRNA-mediated transduction; efforts to accomplish mRNA delivery via liposomes or LNPs to non-muscle site have been less useful owing to limiting in vivo gene delivery efficiency [28,29,30]. This phenomenon represents a major impediment to realizing the full potential of mRNA vectors. For example, the development of vaccines for COVID-19 has highlighted the key functional gains that accrue mucosal immunization [8,31,32,33,34,35]. In this context, there are not currently available liposomes or LNP vectors capable of efficient delivery of mRNA via the mucosal route. On this basis, it has not been possible here-to-fore to accrue the advantages of both mRNA vaccines and mucosal immunization. This recognition provides the driving rationale for our planned future efforts to exploit AdpL-based mRNA delivery for mucosal immunization. Of note in this regard, our historical studies here have confirmed the in vivo capacity of AdpL for mRNA delivery. Further, our studies with AdpL/DNA have clearly demonstrated a capacity for in vivo gene delivery to mucosal epithelium [12,13].

Critical to our overall strategy is the complexing of mRNA to the adenovirus capsid. For proof-of-principle here we employed an early design of adenovirus conjugated to polylysine [10]. This configuration was derived for AdpL-mediated delivery of plasmid DNA and employed here to evaluate the possibilities of mRNA delivery via this approach. In this design the polylysine nucleic acid binding domain is conjugated to the adenovirus capsid utilizing biotin-labelling of adenovirus and combination with streptavidin-polylysine. Of note, there have been substantial technical advances that now feasibilize the derivation of AdpL of greater utility. In the first regard, the advent of “molecular glue” methods (ex. SpyTag/SpyCatcher) now allow directed non-random chemical coupling to derive macromolecular structures [36,37,38,39]. In addition, we have now defined specific capsid locales which can be modified for coupling with such a molecular glue strategy [40]. We are thus now exploring methods to accomplish directed attachments of nucleic acid binding domains to specific adenovirus capsid locales. Further alternative mRNA-binding proteins may offer advantages compared to polylysine. Future designs of the AdpL may thus embody a greatly augmented ability to complex mRNA with less potential to adversely impact the functionality of the adenovirus itself.

## 4. Methods

### 4.1. Production of Retargeted Adenovirus

AdPK4.CMV.GFP was described previously [17]. Seed viruses were serially propagated in HEK293 cells and purified by ultracentrifugation on CsCl gradients according to published protocols [41,42]. Purified viruses were dialyzed against phosphate-buffered saline (PBS) only or using desalting columns (Zeba^TM^ spin desalting columns, 7000 MWCO, Thermo Scientific, Waltham, MA USA) to prepare stocks in an amine-free buffer condition, and stored at −80 °C. Viral particles (vp) were determined by measuring absorbance of the dissociated virus at A260 nm [43].

### 4.2. Preparation and Validation of Biotinylated Adenovirus

To prepare biotinylated virus ([B]-labelled AdPK4) for the AdPK4-STAVpLys-mRNA complexes, EZ-Link^TM^Sulfo-NHS-LC-Biotinylation kit (Thermo Scientific, Waltham, MA USA Cat No. 21435) was used according to the manufacturer’s instructions. Briefly, 1 × 10^11^ vp of purified virus (AdPK4.CMV.GFP) were mixed with Sulfo-NHS-LC-Biotin solution, and excess biotin reagent was removed using a desalting column. Successful biotinylation was confirmed via a HABA assay measuring absorbance (O.D) at A490–500 nm, and stored at −80 °C (Appendix A).

### 4.3. Construction of AdPK4-STAVpLys-mRNA Complexes

AdpK4-STAVpLys-mRNA was complexed in accordance with published methods with slight modification [44]. Briefly, 400 to 800 ng streptavidin-conjugated cationic polylysine (STAVpLys250, 0.1 mg/mL) was first complexed with different amounts of mRNA in 30 µL HBS (20 mM HEPES pH 7.3, 150 mM NaCl) as total equal volumes, and mixed by gently pipetting, then incubated for 20 min at room temperature. This complex was then conjugated with 1 × 10^10^ viral particles of biotinylated adenovirus via the natural high affinity streptavidin-biotin interaction by mixing for 30 min at room temperature [44]. STAVpLys250 reagent was kindly provided by Dr. Wagner’s laboratory, and was validated in their prior studies [10,15,16,44]. mRNA encoding luciferase or GFP was provided by GreenLight Biosciences.

### 4.4. In Vitro Evaluation of Gene Transfer by AdPK4-STAVpLys-mRNA Complexes

To assess in vitro gene transfer mediated by AdPK4-STAVpLys-mRNA complexes, the mRNA encoding luciferase gene was employed. HEK293 or CHO cells were seeded into 96 well plates, and approximately 16 h later infected with 1000 MOI AdPK4-STAVpLys-mRNA (Luc) complexes as determined by absorbance at A260 nm. Forty-eight hours later cells were harvested and expression of luciferase was assessed from whole cell lysates of the indicated samples (AdPK4 only, STAVpLys-mRNA only, non-biotinylated AdPK4-STAVpLys-mRNA complexes and biotinylated AdPK4-STAVpLys-mRNA complexes). Luciferase expression were measured by luciferase assay in accordance with standard protocols (Luciferase assay system, catalog number E1500; Promega, Madison, WI USA). Assay plates were read in a microplate luminometer (Berthold detection system) and analyzed by Graph Pad Prism v8.0c software. In the parallel, infection efficiency was determined using GFP reporter expression through the retargeted adenovirus (AdPK4.CMV.GFP) via fluorescence microscopy. Similarly, to assess co-delivery of multiple mRNAs we utilized an HAdV5 based vector expressing RFP (Ad.CMV.Red). This virus was biotinylated as above and complexed with 0.375 ug each of mRNA encoding luciferase and mRNA encoding GFP. HEK293 cells were seeded, infected and analyzed as described above [42].

### 4.5. In Vivo Evaluation of Gene Transfer by AdPK4-STAVpLys-mRNA Complexes

For in vivo analysis of gene transfer from AdPK4-STAVpLys-mRNA, the 1 × 10^10^ VP complexes were injected intramuscularly (i.m.) into 8 week-old triple immunodeficient NOD/SCID/IL2Rγ (NSG) mice. Two days (48 h) later, the mice were euthanized and sacrificed following approved animal study protocols. Tissues were then harvested. Immunohistochemistry (IHC) was conducted in accordance with our previously published methods as described below [42,45]. All animal experiments were reviewed and approved by the Institutional Animal Care and Use Committee of the Washington University in Saint Louis, School of Medicine (protocol # 20191116).

### 4.6. Immunohistochemical (IHC) Staining Analysis

Harvested mouse organs were fixed with 10% formalin phosphate solution and preserved in 30% sucrose in PBS at 4 °C overnight. The fixed tissues were embedded in NEG50 mounting medium (Thermo Scientific, Waltham, MA USA) then frozen in liquid nitrogen pre-chilled 2-methylbutane. Tissue slide sectioning was performed from frozen tissues using the CryoJane taping system (Leica CM1900). All tissues slides were subjected to IHC analysis with specific primary antibodies (rabbit anti- luciferase [1:200, catalog number ab21176; Abcam, Boston, MA USA]) and using Alexa Fluor 488- or Alexa Fluor 594-conjugated secondary antibodies (1:400, #103-545-155 or #711-585-152; Jackson ImmunoResearch Laboratories, West Grove, PA USA). Images were analyzed using CellSens Dimension imaging software (Olympus Soft Imaging Solutions). All processes were in accordance with previously published protocols [42,45].

## Figures and Tables

**Figure 1 viruses-14-02169-f001:**
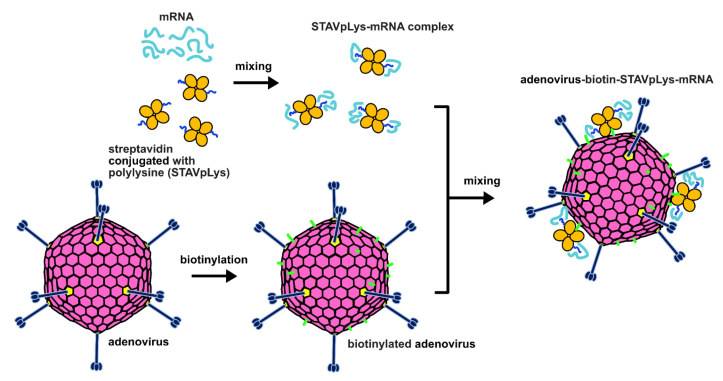
Schematic illustration of a novel chimeric nanosystem for mRNA delivery. In this study, we employed a straight-forward three step approach to prepare AdpL-mRNA complexes. Anionic mRNA is first mixed with cationic streptavidin tagged polylysine to form complexes capable of binding with biotin. In parallel, the adenovirus is biotinylated via chemical labelling of accessible amine groups using the EZ-Link kit. Lastly, the complexed STAVpLys-mRNA is conjugated to the biotinylated virus via simple mixing. This strategy allows for rapid generation of a functional mRNA vector using a pre-prepared adenovirus.

**Figure 2 viruses-14-02169-f002:**
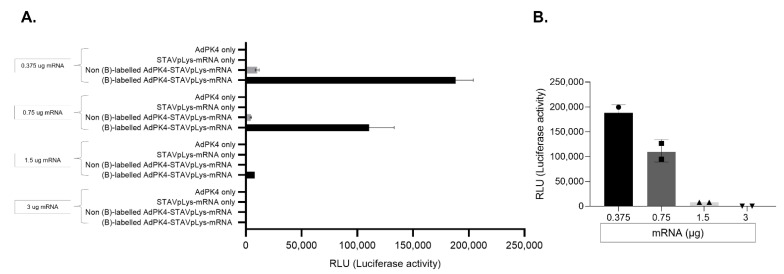
Evaluation of gene transfer efficiency mediated by AdPK4-STAVpLys-mRNA complexes. To evaluate mRNA gene transfer mediated by the AdpL vector, a fixed quantity of biotinylated virus (Appendix A) containing 800 ng StAV conjugated with 880 ng pLys was complexed with different amounts of mRNA (0.375, 0.75, 1.5, or 3 ug) and tested in CHO cells by measuring luciferase gene expression forty-eight hours after virus infection in parallel with controls Average of three technical replicates (**A**). Presentation of the dose-dependence of mRNA gene transfer efficiency. Average of two technical replicates (**B**).

**Figure 3 viruses-14-02169-f003:**
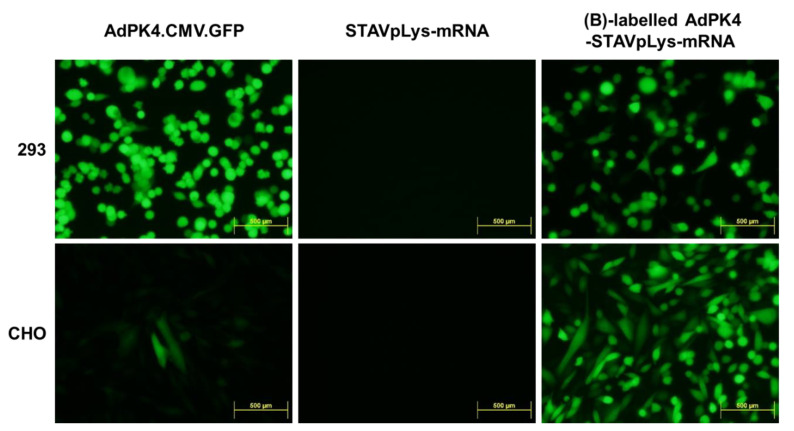
Polylysine enhancement of adenovirus gene transfer. Adenovirus permissive (HEK293) and non-permissive (CHO) cell lines were infected with either AdPK4.CMV.GFP alone, polylysine conjugated mRNA alone (STAVpLys-mRNA) or the AdpL-mRNA conjugate. The AdpL conjugate demonstrated enhanced gene transfer of GFP encoded by the viral DNA to CHO cells, potentially via interaction of polylysine with the cell surface. The fluorescence images were analyzed forty-eight hours after virus infection using fluorescence microscopy.

**Figure 4 viruses-14-02169-f004:**
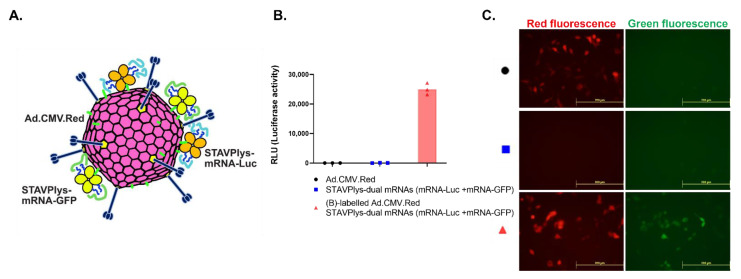
Co-delivery of multiple mRNAs via chimeric nanosystem. To explore the possibility of the simultaneous delivery of two mRNAs (mRNA encoding luciferase and mRNA encoding GFP, 0.375 ug each) via our piggyback approach, in vitro gene transfer was evaluated and confirmed with both gene signals detected (Luciferase activity and GFP expression) in 293 cells. The schema for nanosystem-based piggyback strategy for two mRNAs is shown at the left (**A**) with data in middle [Luciferase activity] (**B**) and in right [Green fluorescence] (**C**).

**Figure 5 viruses-14-02169-f005:**
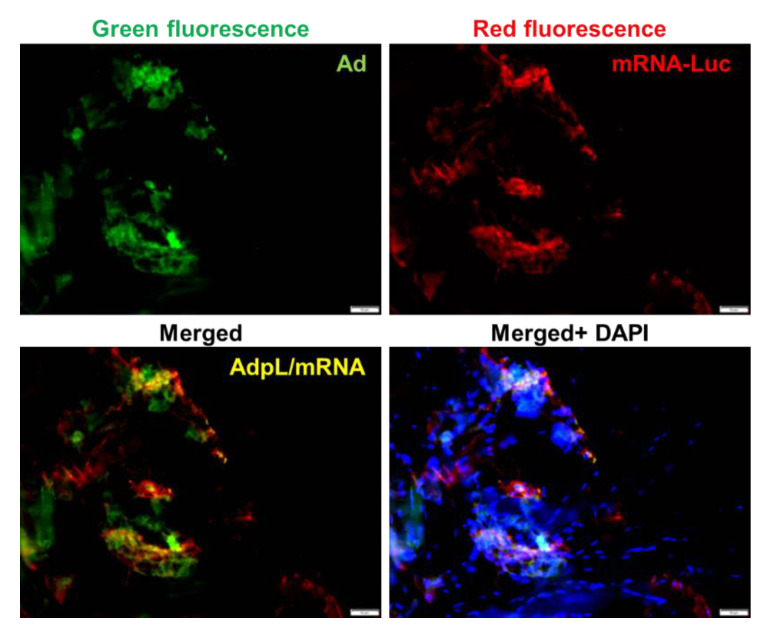
In vivo gene delivery via AdpL-mRNA. AdPK4-STAVpLys-mRNA complexes were administrated via intramuscular (i.m) injection to mice and immunohistochemical staining (IHC) analysis was performed from muscle tissue to evaluate in vivo gene transfer. Gene expression was verified and analyzed by co-localization signals using fluorescence microscopy. Indicated signals are as follows: Green fluorescence: reporter gene via adenoviral vector (AdPK4.CMV.GFP), Red fluorescence: firefly luciferase gene expressed by mRNA delivery using anti-firefly luciferase antibody capture, co-localization represented by yellow in merged. The fluorescence images were taken using epifluorescence microscopy (Olympus America, Center Valley, PA, USA).

**Figure 6 viruses-14-02169-f006:**
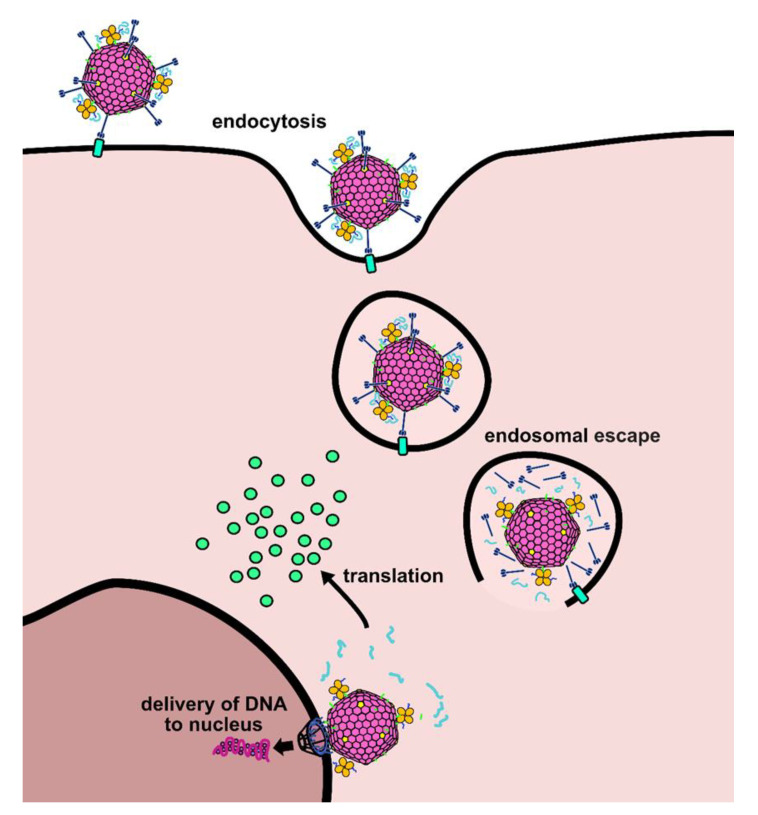
Proposed mechanism for mRNA delivery via AdpL nanosystem. In our proposed model, mRNA delivery is mediated by adenovirus cell entry biology. The AdpL-mRNA conjugate first binds to the target cell via interactions with the fiber protein and is taken up into endosomes. Previously described adenovirus biology results in escape of the conjugate from endosomes, at which point the mRNA may be released from the polylysine and translated into protein in the cytosol. The AdPK4 entry pathway may be different from the well-described HAdV5 pathway, but we hypothesize it will follow a similar mechanism.

## Data Availability

Not applicable.

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
