# Peer review of "A Novel Piggyback Strategy for mRNA Delivery Exploiting Adenovirus Entry Biology"

_viruses, 2022, doi:10.3390/v14102169_

Round 1
Reviewer 1 Report
Myungeun Lee and coworkers report the use of polylysine modified adenovirus particles to deliver mRNA into cells. The authors use a GFP recombinant human adenovirus type 5 (HAdV-5) with the fiber knob of porcine adenovirus type 4, which was shown previously to increase virus tropism. The viral particles were biotinylated and then coupled with a conjugate formed by streptavidin tagged polylysine and a mRNA that encodes luciferase. The modified adenovirus (AdpL) was then used to infect 293 and CHO cells, and expression of luciferase and GFP were evaluated. The AdpL virus was also evaluated in vivo, administered into immunodeficient mice, and expression of luciferase and GFP was evaluated by immunofluorescence in mice organs.
The study is of interest because of the proven potential of AdV as vaccine vectors and the added advantage of a mRNA-delivery vector.
The experimental design is straightforward and the methods and results are presented clearly; however, some aspects, both in terms of the readability and content could be improved:
Major points:
1. The authors claim that the mRNA delivery strategy exploits the entry mechanism of adenovirus. This claim is based on the assumption that the AdpL particles enter the cells through the same pathway that has been described for HAd5. The experiments do show that both the mRNA that is bound to the virus and the viral genome enter the cell, because the luciferase and the GFP proteins, respectively, are produced in the infected cells. However, these experiments cannot be used to infer whether virus entry of the AdpL occurs via the same endocytic pathway as the unmodified virus. Is there evidence that the AdpL virus enters the cells through clathrin (or other)-dependent endosome pathway?
2. The title of the second section of Results, Definition of the molecular basis of adenovirus-polylysine-mediated mRNA delivery does not seem to be justified; rather the experiments in this section describe the experimental parameters that were evaluated to determine the efficiency of mRNA delivery into the cells.
3. Could the authors elaborate on the reasons of the unexpected finding that the higher amounts of mRNA result in decreased efficiency of infection? mRNA delivery? mRNA expression?
4. In the experiments that employ the delivery of two different mRNAs, the percentage of cells that express only one or both the AdV-expressed red fluorescent protein and the mRNA-expressed green fluorescent protein should be determined. From the low quality/resolution fluorescence microscopy images it seems only a small percentage of the cells express both proteins. This may indicate that only a fraction of the AdpL particles is coupled to the mRNA.
5. I recommend improving the quality or presentation of the fluorescence microscopy in figures 4 and 5. For the in vivo experiments the figure may improve including a bright-field image of the cells and a higher amplification or zoom to indicate the cells in the sample.
6. Epifluorescence microscopy cannot be used to determine colocalization of proteins, rather an amplified section of the micrographs in figure 5 could be used to indicate areas where the distribution of the green and red fluorescence signals coincide (yellow).
Minor points:
1. The Methods section describes the use of 1000 MOI AdPK4-STATVpLys-mRNA (Luc) complexes. How was the title of the virus determined?
2. Can the authors briefly describe the choice of the triple immunodeficient NOD/SCID/IL2Rgamma (NSG) mice for the experiments?
3. L25: I would recommend a different word for "feasibilize".
4. When referring to human adenovirus type 5 I recommend using the ICTV accepted nomenclature: HAdV5
5. L103: I recommend rewording: "We tested in CHO (Chinese Hamster Ovary) cells with AdPK4 only,..."
6. L193: To improve clarity and readability I recommend rewording: "Critical to the functionality of the AdpL vector are the functions provided by the adenovirus pursuant to mRNA..."
7. L214: Should read, "The issue of efficiency is of even greater..."
8. L244-245: To improve clarity and readability I recommend rewording: "...with less potential to adversely impact the adenoviral functionalities critical to mRNA trasnport.
Author Response
Reviewer 1
- The authors claim that the mRNA delivery strategy exploits the entry mechanism of adenovirus. This claim is based on the assumption that the AdpL particles enter the cells through the same pathway that has been described for HAd5. The experiments do show that both the mRNA that is bound to the virus and the viral genome enter the cell, because the luciferase and the GFP proteins, respectively, are produced in the infected cells. However, these experiments cannot be used to infer whether virus entry of the AdpL occurs via the same endocytic pathway as the unmodified virus. Is there evidence that the AdpL virus enters the cells through clathrin (or other)-dependent endosome pathway?
The reviewer is correct and we thank them for pointing out this error. We have altered the manuscript text to reflect the speculative nature of the AdPK4 entry pathway.
- The title of the second section of Results, Definition of the molecular basis of adenovirus-polylysine-mediated mRNA deliverydoes not seem to be justified; rather the experiments in this section describe the experimental parameters that were evaluated to determine the efficiency of mRNA delivery into the cells.
We have modified the section title to more accurately describe the content as per the reviewers suggestion.
- Could the authors elaborate on the reasons of the unexpected finding that the higher amounts of mRNA result in decreased efficiency of infection? mRNA delivery? mRNA expression?
Previous work with AdpL and DNA showed that certain ratios of polycation and DNA generated a “toroid” structure which resulted in optimal gene transfer (Wagner et al, PNAS 1991). We hypothesize that a similar effect is occurring here.
- In the experiments that employ the delivery of two different mRNAs, the percentage of cells that express only one or both the AdV-expressed red fluorescent protein and the mRNA-expressed green fluorescent protein should be determined. From the low quality/resolution fluorescence microscopy images it seems only a small percentage of the cells express both proteins. This may indicate that only a fraction of the AdpL particles is coupled to the mRNA.
We thank the reviewer for their suggestion. Due to the differences between gene expression from a viral backbone and gene expression from an exogenous mRNA, it is difficult to quantitatively compare the two reporters. We have clarified the quantitative nature of the images in our revised text.
- I recommend improving the quality or presentation of the fluorescence microscopy in figures 4 and 5. For the in vivo experiments the figure may improve including a bright-field image of the cells and a higher amplification or zoom to indicate the cells in the sample.
We cannot at present repeat this study but we can address the reviewer’s point. The main point of this study is to show the AdpL can direct in vivo gene transfer via the mRNA-encoded reporter. The co-localization aspect was solely to track the vector. This latter point is subservient to our main intent. We clarify this point in the revised manuscript
- Epifluorescence microscopy cannot be used to determine colocalization of proteins, rather an amplified section of the micrographs in figure 5 could be used to indicate areas where the distribution of the green and red fluorescence signals coincide (yellow).
The response here is as noted in point #5.
Minor points:
- The Methods section describes the use of 1000 MOI AdPK4-STATVpLys-mRNA (Luc) complexes. How was the title of the virus determined?
We thank the reviewer for their attention to detail and have clarified that virus titer was determined via absorbance at A260 nm.
- Can the authors briefly describe the choice of the triple immunodeficient NOD/SCID/IL2Rgamma (NSG) mice for the experiments?
In our initial studies we wished to avoid any possible complications from immune reactions to the delivered AdpL-mRNA conjugate, which informed our choice of mouse strain. In future studies we aim to transition our system to an immune competent model for vaccine studies.
- L25: I would recommend a different word for "feasibilize".
We have altered the phrasing as per the reviewer’s request.
- When referring to human adenovirus type 5 I recommend using the ICTV accepted nomenclature: HAdV5
We thank the reviewer for this clarification and have modified the text as suggested.
- L103: I recommend rewording: "We tested in CHO (Chinese Hamster Ovary) cells with AdPK4 only,..."
We have modified the text as recommended.
- L193: To improve clarity and readability I recommend rewording: "Critical to the functionality of the AdpL vector are the functions provided by the adenovirus pursuant to mRNA..."
The requested sentence has been reworked as per the reviewer’s comment.
- L214: Should read, "The issue of efficiency is of even greater..."
We thank the reviewer for their comment and have modified the text.
- L244-245: To improve clarity and readability I recommend rewording: "...with less potential to adversely impact the adenoviral functionalities critical to mRNA trasnport.
We have rewritten the text as requested.
Reviewer 2 Report
In the manuscript titled “A Novel Piggyback Strategy for mRNA Delivery Exploiting 2 Adenovirus Entry Biology”, the authors used a modified adenoviral vector to facilitate transport of mRNA by piggybacking on the capsid exterior of the adenovirus. The following points could be considered to improve quality of the manuscript:
1) In Figure 2, some luciferase signal can be observed in Non-Biotin-labelled AdPK4-STAV4Lys-mRNA. Can the authors discuss that in the results section?
2) In Figure 2, what the lowest concentration of mRNA that will yield a luciferase signal, do the authors expect a higher signal when using mRNA less than 0.375ug?
3) In figure 2, was the experiment performed using three biological replicates? If yes, can the author include that in the text?
4) If using more than 800ng StAV and 880ng pLys, do the authors expect the deliver more mRNA?
5) Is there any effect of the AdpL gene transfer on cell viability?
6) Have the authors considered using some other method to quantify gene expression? Like qPCR or western blotting, this would be useful when using mRNA coding for genes other than reporters.
7) In Figure S1, what is concentration of Non (B) labelled virus used for this experiment, 1X,2X or 4X, please mention in the text.
8) Do the authors anticipate an effect of virion concentration on gene delivery, here only one concentration is used.
9) In Figure 4, GFP fluorescence is not quantitative and indicative of the amount of mRNA expression, can the authors provide any flow cytometry measurement for the same?
10) In figure 4, what concentration of both GFP and luciferase mRNA is being used here, please mention in the text.
11) Do the authors anticipate the effect of the length of mRNA have an impact on gene delivery, if yes, can they use different size for the same? What is the size of the mRNAs used here?
Author Response
Reviewer 2
1) In Figure 2, some luciferase signal can be observed in Non-Biotin-labelled AdPK4-STAV4Lys-mRNA. Can the authors discuss that in the results section?
In our previous work we demonstrated that Ad uptake into cells could induce a “by-stander” effect leading to increased uptake of other molecules (Curiel et al, PNAS 1991). We hypothesize a similar effect may be at play here and have included this hypothesis in our revised manuscript.
2) In Figure 2, what the lowest concentration of mRNA that will yield a luciferase signal, do the authors expect a higher signal when using mRNA less than 0.375ug?
We thank the reviewer for their interest – at this time we have not studied the use of lower amounts of mRNA. This will be included in our future studies.
3) In figure 2, was the experiment performed using three biological replicates? If yes, can the author include that in the text?
Figure 2A presents the average of three technical replicates, while Figure 2B presents the average of two technical replicates. We have clarified this in the revised text.
4) If using more than 800ng StAV and 880ng pLys, do the authors expect the deliver more mRNA?
We thank the reviewer for their comment. We have not yet investigated the effects of altering each parameter of our system, but will include this research in our ongoing work with this system.
5) Is there any effect of the AdpL gene transfer on cell viability?
We have not launched a formal investigation into this issue, but anecdotally have not seen any impacts on cell viability at the MOIs used.
6) Have the authors considered using some other method to quantify gene expression? Like qPCR or western blotting, this would be useful when using mRNA coding for genes other than reporters.
We thank the reviewer for this suggestion – understanding and quantifying the types of mRNA that can be delivered by the AdpL system is an area of ongoing research in our laboratory.
7) In Figure S1, what is concentration of Non (B) labelled virus used for this experiment, 1X,2X or 4X, please mention in the text.
1X, 2X and 4X are equivalent to 1E10, 2E10 and 4E10 viral particles, respectively. We have clarified this information in the revised manuscript.
8) Do the authors anticipate an effect of virion concentration on gene delivery, here only one concentration is used.
We thank the reviewer for their interest - in this study, we utilized a vector dose routinely established in our lab for reference purposes. In further studies we will include optimization of more parameters including the viral particle concentration.
9) In Figure 4, GFP fluorescence is not quantitative and indicative of the amount of mRNA expression, can the authors provide any flow cytometry measurement for the same?
We appreciate the reviewer’s attention to detail - at this juncture our study is primarily qualitative, which we have clarified in the manuscript.
10) In figure 4, what concentration of both GFP and luciferase mRNA is being used here, please mention in the text.
Each mRNA was used at a total amount of 0.375ug in this study – we have altered the figure to more accurately reflect this.
11) Do the authors anticipate the effect of the length of mRNA have an impact on gene delivery, if yes, can they use different size for the same? What is the size of the mRNAs used here?
We thank the reviewer for their comment. We are actively investigating which lengths and sequences of mRNA can be successfully delivered by the AdpL system.